# Osseointegration of Zirconia Implants after UV-Light or Cold Atmospheric Plasma Surface Treatment In Vivo

**DOI:** 10.3390/ma15020496

**Published:** 2022-01-10

**Authors:** Lisa Krautwald, Ralf Smeets, Carolin Stolzer, Rico Rutkowski, Linna Guo, Aline Reitmeier, Martin Gosau, Anders Henningsen

**Affiliations:** 1Division “Regenerative Orofacial Medicine”, University Medical Center Hamburg-Eppendorf, Martinistrasse 52, 20246 Hamburg, Germany; l.krautwald@uke.de (L.K.); r.smeets@uke.de (R.S.); guolinna001@gmail.com (L.G.); m.gosau@uke.de (M.G.); 2Department of Oral and Maxillofacial Surgery, University Medical Center Hamburg-Eppendorf, Martinistrasse 52, 20246 Hamburg, Germany; c.stolzer@uke.de (C.S.); r.rutkowski@uke.de (R.R.); 3Department of Stomatology, The Second Xiangya Hospital, Central South University, Changsha 410011, China; 4Department of Laboratory Animal Science, University Medical Center Hamburg-Eppendorf, Martinistrasse 52, 20246 Hamburg, Germany; a.reitmeier@uke.de; 5Private Practice ELBE MKG, Suelldorfer Kirchenweg 1A, 22587 Hamburg, Germany

**Keywords:** dental implants zirconia, osseointegration, cold atmospheric plasma, UV light

## Abstract

The influence of UV light and non-thermal plasma on the osseointegration of yttria-stabilized zirconia implants (Y-TZP) comparing the two methods is unclear. The aim of this study was to show the influence of these methods on the osseointegration of dental zirconia implants in an animal model. A total of 54 implants were either untreated, treated with UV light (UV), or non-thermal oxygen plasma for 12 min and inserted into the parietal bones of six domestic pigs. The animals were sacrificed after a healing interval of two, four, and nine weeks. The degree of osseointegration was determined using histomorphometric determination of bone-to-implant contact values (BIC) and the bone-to-implant contact values within the retentive parts of the implants (BAFO). BIC values decreased in all groups after four weeks of healing and re-increased after nine weeks in all groups. BAFO increased significantly over time in all groups. However, there were no statistically significant differences in BIC and BAFO values between the control group and the test groups and over time. Clinical studies may follow to confirm the influence of cold plasma and UV light on the healing and survival of zirconia implants.

## 1. Introduction

Dental implants have evolved a profound method to replace missing teeth. Osseointegration, which describes the direct structural and functional contact between the human bone and the implant surface, is one key factor for long-term success and survival of dental implants [1]. Additionally to functionality and stability, esthetics have become increasingly important in recent years following the introduction of zirconia as ceramic implant material [2]. Zirconia is highly biocompatible and possess good stability properties. However, some studies revealed long-term outcomes may be inferior to those of comparable studies that investigated titanium implants [3,4,5].

During implant healing, the surface is primarily covered by woven bone, which is later replaced by realigned lamellar bone (remodeling). At the end of this remodeling phase, usually 60–70% of the implant surface is covered by bone that can be measured histomorphometrically using the bone-to-implant contact [6]. BIC is the most common measuring method in order to assess osseointegration in vivo [7]. The BIC typically ranges between 65% and 73% and does usually not reach the ideal 100% using modern implant systems [8]. Another term to assess osseointegration in vivo is bone area fraction occupancy (BAFO), which are the areas occupied by bone between the implants’ threads subtracted from the total area between the threads (“healing chambers”) reported in percentage values according to Leonard et al. [9].

Generally, the physical, topographical, and chemical surface properties of implants determine their ability to osseointegrate. An ideal physico-chemical composition or surface topography of dental implants is currently unknown and chemical surface changes as well as adjustments to the topography in the nanometer range are the subject of recent research [2]. After manufacturing, implants are gamma-sterilized, but usually, the package is permeable to air. Comparable to titanium implants, biological aging may be a reason for a reduced osseointegrative capacity of zirconia implants. Carbon compounds that deposit on the surface during storage can cause hydrophobization and decreased protein-binding capacity [10]. Additionally, the increasing age of the population and the use of bone metabolism-interacting drugs like bisphosphonates, glucocorticoids, and cytostatics may lead to compromised osseointegration of dental implants by reducing its biological activity [11,12]. Studies showed that surface treatment using ultraviolet (UV) light or cold atmospheric plasma (CAP) may be capable of reducing hydrophobic layers of hydrocarbons and carbonaceous species and may increase hydrophilization, cell adhesion, and proliferation on implant surfaces that are made of titanium or zirconia [13,14,15,16,17,18,19]. Studies revealed that a surface treatment of zirconia using short-wave UV light may have a positive effect on osseointegration of dental implants [18]. Irradiation with short-wave light breaks molecular connections and an increased surface polarity leading to an increased reactivity of the treated surfaces [20]. Cold atmospheric plasma (CAP) is already used for disinfection and sterilization of medical products. Similar to the effects of UV-light, energy is administered to the surface, electrons are released and reactive particles are created [21]. The increased reactivity of the surface leads to a surface hydrophilization and may increase cell attachment and the viability of osteoblasts without changing the physical surface properties [22]. However, a comparison of the effects of UV-light and CAP on the osseointegration of zirconia implants in vivo has not been performed yet.

The aim of this study was to determine the influence of UV-light and cold plasma on osseointegration of yttria-stabilized zirconia implants in vivo. The hypothesis was that the use of UV-light or CAP on these implants prior to implant placement can increase the histomorphometrical osseointegration of the implants measured by BIC and BAFO in a landrace pig model.

## 2. Materials and Methods

### 2.1. Implants

One-piece whiteSKY zirconia implants (Bredent GmbH & Co. KG, Senden, Germany), 4 mm in diameter and 8 mm in length, were used in this study (Figure 1). The implant design is considered to be particularly stable because there is a screw connection to fix an abutment on the implant. The manufacturer specifies a flexural strength of 1250 MPa +/− 120 MPa, a modulus of elasticity of 200 GPa and a fracture toughness of 6–8 MPa/m for these implants. In total, 54 implants were used in this study, of which 18 implants used non-treated as a control group. The surface structure was investigated using a scanning electron microscope (Zeiss Crossbeam 340, Carl Zeiss AG, Oberkochen, Germany).

### 2.2. Surface Treatments

In total, 36 implants were treated with UV-light or cold atmospheric plasma. Immediately before insertion, the implants of one experimental group (18 implants) were placed in a UV-light oven for 12 min that generates UV-light with an intensity of 0.15 mW/cm^2^ (λ = 253.7 nm). The second group of 18 implants were placed in a plasma reactor (Yocto III, Diener Electronic GmbH, Ebhausen, Germany) for 12 min using cold atmospheric argon plasma (generator frequency 100 kHz, input power 24 W, system pressure 1 mbar, gas flow rate 1.25 sccm, gas purity > 99.5%) immediately before insertion (Figure 2). The protocols are established and were used in several previous studies [23,24,25].

### 2.3. Animals

The animal study was approved by the Authority of Health and Consumer Protection of Hamburg (approval number 040/2018) and it was conducted following the ARRIVE guidelines according to the German legislation on protection of animals and the NIH guidelines for the care and use of laboratory animals. Sample size (six animals, nine implants per animal, and three different time points) was calculated with a power of 0.89 using a biological relevant difference of 15% and an effect size of 1.22 (Cohen’s f). The implant placements in the frontal calvarial bone of six juvenile female German landrace pigs took place from June 2019 to August 2019 at the animal facility of the University Medical Center Hamburg-Eppendorf. A frontal calvarial bone model was used because it enables to place up to 9 implants using only one surgical approach in a short time in order to reduce the stress on the animals. The animals (between 55.5 and 58.5 kg at start of the investigation) were kept together in adjacent lying sheds and were fed with pellet food and water. After two weeks of acclimatization at the animal facility, the operations were performed.

### 2.4. Surgical Procedures

Anesthesia was maintained by 10 mg/kg Ketamine (HVW Bremer Pharma, Warburg, Germany) and 4 mg/kg Azaperon (Jansen, Bad Homburg, Germany). After endotracheal intubation (standard tube with a diameter of 6–7 mm from Portex, Kent, UK), anesthesia was continued by inhalation with Isoflurane (Forene, Abott GmbH, Wiesbaden, Germany) in a weight-adapted manner. The animals received a perioperative antibiotic prophylaxis with Enrofloxacin (10% Baytril, Bayer, Leverkusen, Germany).

After placing the animals in a prone position and local disinfection (Betaisadona, Braun, Melsungen, Germany), the frontal calvarial bone was exposed using a 10 cm long vertical incision after local infiltration anesthesia with UDS forte (Hoechst GmbH, Frankfurt, Germany). Nine implants were randomly placed into the ossa calvaria. Each animal received 3 untreated, 3 UV-light treated, and 3 CAP-treated implants. The pilot drilling was carried out under water cooling and it was prepared step-by-step in accordance to the manufacturer’s protocol. The implants were inserted primary stabile with an insertion torque of 45 N/cm. Finally, wounds were closed in multilayers using Vicryl 2-0 (Ethicon Co., Norderstedt, Germany) and the animals received 4 mg/kg Rimadyl (Zoetis, Berlin, Germany) intravenously for analgesia. The duration of each operation was between 40 and 60 min. Two animals were euthanized after two weeks (group 1), another two animals after four weeks (group 2), and the remaining two animals after nine weeks (group 3). The whole calvarial bone was removed and placed into aqueous solution of 4% formaldehyde.

### 2.5. Histomorphometrical Preparation and Analysis

For histomorphometrical analysis, each of the nine implants containing the surrounding bone tissue was sawed out using a diamond band saw (EXAKT 310 CP with diamond separating band 0.3 mm D64, Exakt Apparatebau, Norderstedt, Germany), resulting in nine implant-bone-blocks. The specimens were processed to conventional X-ray (Faxitron X-ray, Faxitron Xray Corp., Wheeling, IL, USA) following preservation in aqueous solution of formaldehyde (3.5%). Afterwards, the specimens were dehydrated using an ascending ethanol solution (Geyer GmbH, Hamburg, Germany) and embedded in resin. Parallel sample blocks (thickness 300 µm) were cut off using a diamond band saw (EXAKT cut-off grinding system Makro 310 CP with EXAKT diamond cutting tape 0.1 mm D64 310, EXAKT Apparatebau, Norderstedt, Germany) and thinned and polished using a micro-grinding system to finally 80–100 µm thickness (grain size 1200/2500/4000 SiliconCarbide Paper, Allied High Tech Products Inc., Rancho Dominguez, CA, USA). The histological slices were stained using toluidine blue (Kohler Chemie GmbH, Bensheim, Germany). Microscope images were created using an Axio Scope A1 (Carl Zeiss AG Oberkochen, Germany) and measurement were performed using OsteoMeasure XP software (release 3.2.1.1., Osteometrics, Atlanta, GA, USA). First, the entire implant surface surrounded by bone was measured. To determine the BIC (bone-to-implant contact), the parts of the implants surface covered with mineralized bone tissue was placed in relation to the total surface giving the percentage of the surface that is successfully osseointegrated (Figure 3). To determine the BAFO (Bone area fraction occupancy), the total area between the threads was measured following indication of all areas occupied by bone in these “healing chambers”. The percentage of area occupied by bone in mm^2^ in relation to the total area in mm^2^ gave the BAFO (Figure 4) [9]. Measurements were performed by two independent researchers.

Implants showing BIC values below 20 were considered to be failures. These zirconia implants are not considered to be osseointegrated and were removed from the evaluation.

### 2.6. Statistical Analysis

Inter-observer reliability was assessed with an intraclass correlation coefficient for consistency and absolute agreement of the data in a two-way mixed model. A generalized linear mixed model (GLMM) was used for statistical analysis using SAS 9.4 (SAS Institute Inc., Cary, NC, USA). The treatment method and the healing time were modelled as fixed effects in the mixed model. The individual characteristics of the pigs were adopted as random effects using a compound symmetry covariance structure. The final group comparison was based on least-square-means estimators, which were constructed using the respective parameter estimators. *p*-values < 0.05 were considered to be statistically significant.

## 3. Results

### 3.1. Scanning Electron Microscopy

Scanning electron microscopy revealed a micro-rough surface in the region of the threads and a polished surface in the upper part of the implants starting at the implant neck for optimized soft tissue adhesion (Figure 5).

According to the manufacturer, the micro-rough surface in the region of the threads is achieved using sand-blasting (Figure 6).

### 3.2. Animal Evaluation

All operations were performed without complications. The postoperative course was uneventful without wound healing disorders and dehiscences. All animals gained weight and they behaved normally. After two weeks the average weight in all animals was 67.5 kg, after four weeks the average weight was 82 kg, and after 9 weeks it was 99.5 kg.

### 3.3. Macroscopic and Radiological Evaluation

After removal, all calvariae were macroscopically intact with no signs of infection or osteolysis (Figure 7).

Bone was not present around implant number 45 in animal number 5 (UV-light group, nine weeks healing time), which was surrounded by connective tissue. The remaining implants were clinically successfully osseointegrated. Seventy-four per cent (*n* = 39) of the implants were accidentally partially inserted into the sinus due to the limited vertical amount of bone in the pigs. However, all implants were apically covered by mucosa without any signs of healing disorders. After a healing time of nine weeks, four implants were coronally overgrown by bone (two untreated implants and two UV-light treated implants).

### 3.4. Histomorphometrical Evaluation

The results of the reliability tests indicated a high inter-observer reliability for the histomorphometrical measurements regarding consistency (*p* = 0.89, 95%-CI: 0.85; 0.93) and absolute agreement (*p* = 0.92, 95%-CI: 0.87; 0.95) of the data. Exemplary radiographical and microscope images are shown in Figure 8 and Figure 9.

BIC values ranged between 21.9 and 80.1% (Figure 10). Mean values decreased after four weeks of healing in all groups, but rose again after nine weeks of healing in all groups. However, the highest BIC values were achieved in the UV-light group (60% after two weeks of healing, Table 1).

Generally, mean BAFO values ranged between 37.2% after two weeks of healing and 59.7% after nine weeks of healing. In contrast to BIC results, mean values increased continuously until nine weeks of healing except in the CAP group (Figure 11). However, the highest BAFO values were achieved in the UV-light group (60.9% after two weeks of healing, Table 2).

Joining the results of BIC and BAFO, values in all groups were highest after nine weeks of healing, indicating the osseointegration over time. However, the differences regarding BIC and BAFO values between the groups nor in the different groups over time in the groups were not statistically significant (*p* < 0.05) (Table 3 and Table 4).

Table 5 summarizes the results of the implant that were characterized as failures. Generally, number of failed implants decreased after nine weeks of healing (Table 5). However, a systematic reason for failure could not be determined.

## 4. Discussion

Surface functionalization by UV-light and CAP treatment are able to create hydrophilic implant surfaces and may improve osteoblastic adhesion [10]. Biological aging caused by an accumulation of carbon remnants on the implants’ surfaces since manufacturing may be reversed using either UV-light or CAP and several studies have already confirmed these effects on the implants’ surfaces [14,20,23,24,25,26,27,28,29,30]. Modifying zirconia surfaces is more difficult and challenging than metallic surfaces [5]. Additionally, due to its characteristics, zirconia is more hydrophobic than titanium [24]. Therefore, an activation of the surface could be beneficial to establish a rapid and reliable osseointegration. In this study, treatment of zirconia implants using CAP and UV-light initially higher BIC values after two and four weeks of healing. However, differences between the groups were not significant. BAFO values increased over time in the all groups without statistically significant differences between the groups over time. However, after nine weeks of healing, BAFO values of untreated implants were 3.3% higher compared to the UV-light group and were 5.7% higher than in the CAP group. Compared with previous studies investigating the influence of surface treatment on the osseointegration of zirconia implants, the present in vivo study did not show any significant improvements of the histomorphometric healing of zirconia implants.

In this study, the landrace pig was chosen, which is an established model to examine osseointegration. The ratio between trabecular and cortical bone is comparable to human bone [25,31,32]. However, this animal model may be a limitation. The juvenile animals that were used grew very fast, which was reflected in the weight gaining, which means that a higher bone turnover and growth must be assumed compared to humans. A high turnover may also be the reason that there were no statistically significant differences on BIC and BAFO between treated and non-treated implants in this study as well as the reason for the missing increase of BIC over time by reducing the bond between bone and the implants’ surface. The calvarial bone model that was used may also be a limitation due to the reduced vertical amount of bone in the juvenile animals that led to a partial intrasinusoidal placement of implants. However, all of them were apically covered by mucosa without any signs of healing disorders. Although initial osseointegration in the calvarial bone should not be different compared to the jawbones, long-term osseointegration may be varying considering the load of masticatory forces. Treatment using UV-light and cold atmospheric plasma was performed for 12 min in this study. It remains unclear whether longer exposure to zirconia surfaces may achieve higher effects. Studies showed that a longer treatment interval may lead to increased hydrophilicity and improved osseointegration [26,33]. High standard deviation values were determined, especially after two weeks of osseointegration of implants. The low number of animals and implants may be possible reasons. However, another reason may be that the process of osseointegration may be different compared to the initial osseointegration processes of titanium implants. Some studies even described decreased BIC values when comparing osseointegration of titanium and zirconia implants [34,35]. The use of a sealed system generating CAP may lead to a limitation when comparing the results of this study to other studies, since most often open atmospheric plasma devices are used in comparable studies. An explanation for the high standard deviation may be the small number of only six animals. As a result, a greater spread of the results may be expected. To solve this problem, an attempt was made by increasing the number of implants per animal to nine implants each. Another important point to consider is the animals’ high growth rate.

One study compared the influence of UV-light and CAP treatment on the surface structure, cell adhesion, and cytocompatibility of zirconia in vitro [25]. Both surface treatments significantly reduced the amount of organic material on zirconia. There was an increase in hydrophilicity and improved conditions for osteoblasts. Further, functionalization using CAP led to higher cell proliferation and cell attachment compared to UV-light. In this in vivo study, plasma treatment did not significantly improve BIC values after a four-week healing period. Even after nine weeks of healing, however, there was no advantage for the CAP treatment on zirconia with regard to the contact values between bone and implant. It is possible that the surface of the zirconia implants that were used (white SKY, Bredent GmbH & Co. KG) may be improved compared to other zirconia implants and that the effects of CAP or UV-light may be reduced. Another explanation could be that the time of implant placement since manufacturing was too short to get the implants’ surfaces saturated with carbon compounds. The animal model could also be partially responsible for the absent effects of UV-light or CAP because juvenile landrace pigs were used that showed distinct growth effects and gained 68% body weight during only nine weeks of keeping.

However, several in vivo-studies showed positive effects after using UV-light on the osseointegration of zirconia implants. Brezavscek et al. treated smooth and rough zirconia disks and cylindrical implants with UV light for 15 min and inserted them into rat femurs. After two and four weeks, the proportion of bone-to-implant contacts had increased by three to seven times (smooth surfaces) and 1.4–1.7 times (rough surfaces) compared to untreated surfaces. The formation of the bone-implant contacts accelerated and the bond was significantly stronger [36]. Compared to the present study, the irradiation time was 3 min longer at 15 min. The emission of UV light as a spectral mixture at λ = 360 nm and λ = 250 nm was comparable to the present study.

Tuna et al. examined the effect of UV-light on smooth and roughened surfaces made of zirconia on the reaction of alveolar human bone and its osteoblasts. A 15 W bactericidal UV lamp was used and the zirconia discs were irradiated for 48 h with a mixed spectrum between λ = 360 nm and λ = 250 nm. After 24 h, there was a significantly higher number of osteoblasts on the surfaces treated with UV-light and after three weeks, there was significantly more mineralized bone on the UV-treated zirconia surfaces than on the untreated surfaces. The authors concluded that the treatment with UV-light transforms the zirconia surfaces from hydrophobic to hydrophilic which may result in faster healing and increased BIC values [15].

Improved cell reaction on zirconia surfaces after various plasma treatments has also already been reported. Zheng et al. showed an increased cell density after non-thermal plasma treatment using helium as carrier gas [16]. For this purpose, zirconia disks were treated with CAP for 30, 60, or 90 s and the biocompatibility of human gingiva fibroblasts was examined. A 60 s treatment increased the density of fibroblasts on the disks. After 30 and 60 s of treatment, the expression of attachment-related genes was significantly higher in the test groups compared to the controls. After the plasma treatment, the surface wettability increased due to the presence of hydroxide and the biological behavior of the human gingival fibroblasts improved. After a cultivation time of 90 s, the expression of integrins decreased again. The authors believe that atmospheric helium pressure plasma has an early and brief effect on the zirconia disks. Improved cell behavior and a possible improved connection between zirconium oxide and bone tissue was described. In this study, no significant improvement in the BIC and BAFO values was noticed.

Watanabe et al. showed that acid etching, non-thermal plasma and UV-light treatments turned zirconia surfaces hydrophilic and improved the adhesion of osteoblast-like cells. The authors used tetragonal zirconia discs that were sandblasted and etched with hydrofluoric acid. Roughened zirconium oxide surfaces were created. Some of the samples were stored in distilled water for one day, a second part was treated with oxygen plasma for 10 min and a third part was irradiated with UV light for 2 h (using a wavelength between 185 and 365 nm) [27]. The results showed that cell attachment could be improved. However, compared to the present study, the zirconium oxide disks were treated using a significantly longer period of 2 h. This may also be one reason that no significant differences were shown in the histomorphometric analysis.

In another study, a 10 s treatment using non-thermal plasma increased the microtensile bond strength of zirconia in vitro [37]. Zirconia disks (Y-TZP) were loaded with 1 mm/min with 100 N until failure the cross-sectional areas of these samples were measured and the MTBS values (microtensile bond strength) in MPa were calculated showing a clear increase in the test groups. Shon et al. placed PIM (powder-injection molded) zirconia implants in the tibiae of 25 rabbits and were able to show that treatment with helium plasma resulted in a significant increase in BIC values [38]. The rabbits received four types of implants. There was a group with PIM zirconia implants, roughened PIM zirconia implants, plasma-treated PIM zirconia implants, and plasma-treated roughened PIM zirconia implants. The static contact angle of the implant types was evaluated by measurement with a contact angle analyzer using the sessile drop technique. The plasma-treated implants had a contact angle of less than 1°, which suggests superhydrophilicity. A removal torque test was performed, which was influenced by the surface roughness, but not by the plasma treatment. On the other hand, the plasma-treated implants showed a significantly higher BIC than the untreated implants. The result was that the plasma treatment seems to significantly increase the hydrophilicity without changing the surface parameters of the implants. In the present study, implants with only one surface were used. However, although removal torque was not tested, there were also no statistically significant differences comparing BIC and BAFO values of control and test groups.

Hashim et al. conducted a systematic review in 2019 containing 14 articles and reported an overall survival rate of 92% for one- and two-part zirconia implants after one year [39]. Roehling et al. described the one- and two-year survival rates as 98.3% and 97.2%, respectively, after analyzing a total of 18 studies [40]. In the present study, the BIC value of a total of 12 implants was below 20, which means that they were not osseointegrated and therefore had been declared as failures giving a rate of osseointegration of 78%, which is lower compared to the studies mentioned above. However, it could be seen that the rate of implants presenting BIC values below 20% decreased over time. It may be possible that dental implants made of zirconia may need a longer healing time.

Implant loss is most likely caused by impaired osseointegration in the early phase or by a change in the contact between the implant and bone tissue mainly caused by inflammatory reactions on the bone-to-implant interface [41,42,43]. The aging of society is associated with a decrease in bone activity being a risk for successful osseointegration of dental implants [11]. Another reason for impaired osseointegration might be the use of bone-interacting drugs such as glucocorticoids or bisphosphonates [12].

Therefore, it is important to find methods for faster and safer osseointegration of implants. These should be easy to integrate into everyday clinical practice and should be able to be used with lasting effects without major risks for the patient.

In pre-clinical studies, UV-light and CAP were able to improve adhesion of osteogenic cells on zirconia implant surfaces and may increase the speed of osseointegration. Both methods can be performed well in a short and clinically practicable period of time. This study showed that surface functionalization using cold atmospheric plasma and UV-light may be able to increase osseointegration in the crucial first period following implant placement. A positive effect on osseointegration in the critical early phase of bone healing would be important, especially in elder patients or patients taking bone metabolism interacting drugs. However, controlled clinical studies are needed to investigate the determined effects, especially in these patients.

## 5. Conclusions

The aim of this study was to determine the influence of UV-light and cold atmospheric plasma (CAP) on osseointegration of yttria-stabilized zirconia implants in a pig model. Generally, bone-to-implant contacts in groups treated with UV-light and CAP were higher after two and four weeks of healing, but without statistically significant differences. Bone area fraction occupancy (BAFO) increased over time in all groups without statistically significant differences. Twelve implants showed BIC values below 20% and were rated as failures. However, the rate of implants presenting such low BIC values decreased over time, assuming longer healing times for zirconia implants compared to titanium implants. However, zirconia implants may achieve clinical and histomorphometrical osseointegration comparable to titanium implants with improved biocompatibility and esthetics, but modifying the surface with the aim of even increased cell attachment appears to be challenging. Controlled clinical studies are needed to confirm the influence of cold atmospheric plasma or UV-light on the success and survival of dental implants.

## Figures and Tables

**Figure 1 materials-15-00496-f001:**
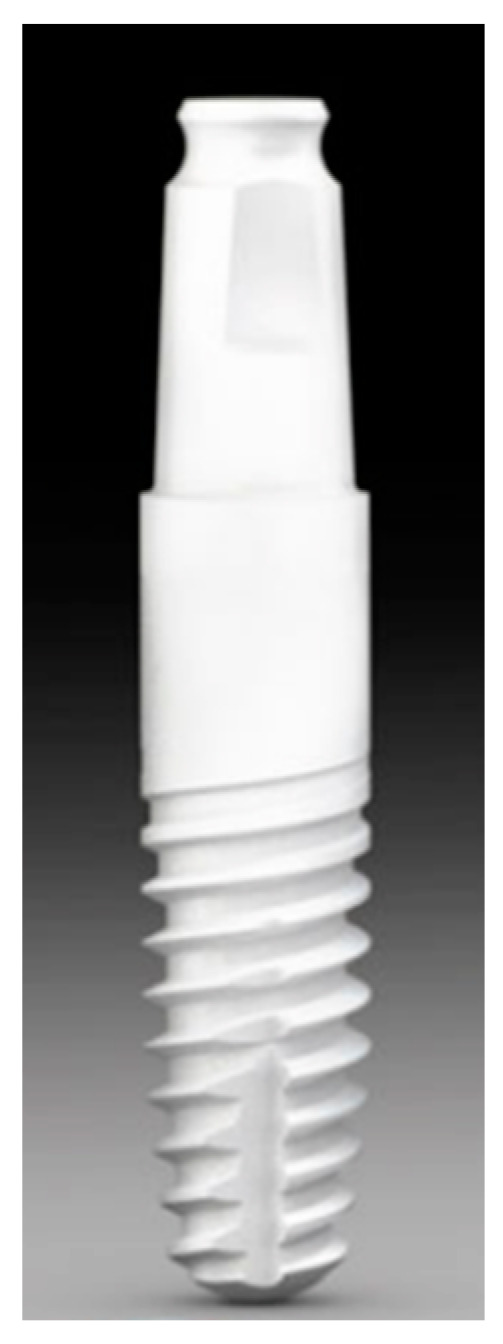
whiteSKY zirconia implant (Bredent GmbH & Co. KG) Adapted from: https://www.bredent.co.uk/products/sky-implants (accessed on 15 October 2021).

**Figure 2 materials-15-00496-f002:**
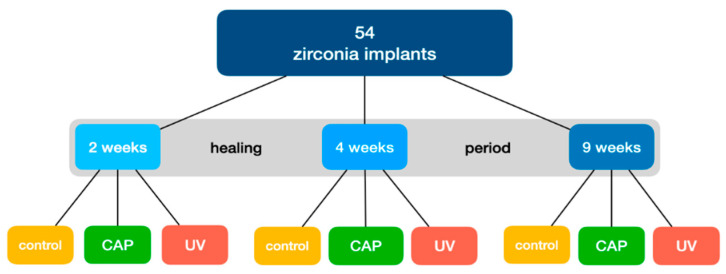
Study flowchart.

**Figure 3 materials-15-00496-f003:**
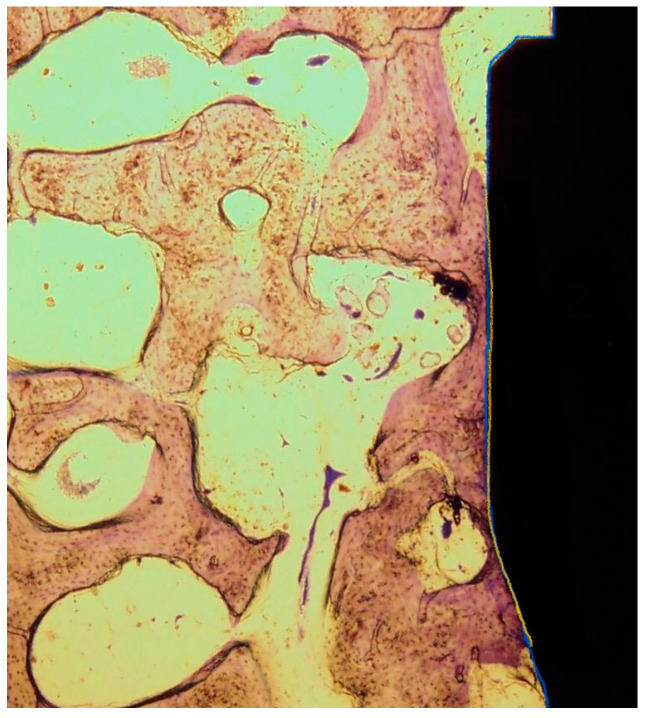
Histomorphometric evaluation of the bone-to-implant-contact (BIC) (blue = entire implant surface, yellow/blue = bone-covered implant surface).

**Figure 4 materials-15-00496-f004:**
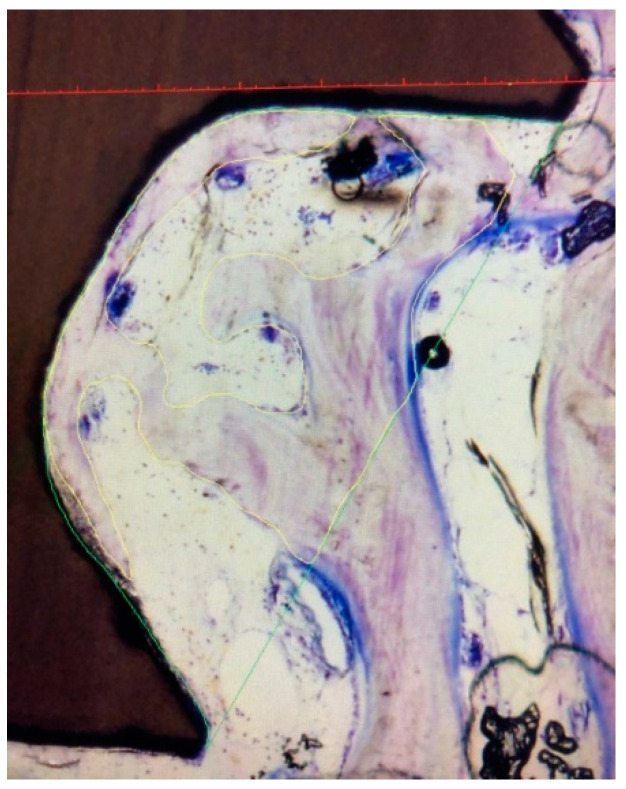
Determination of the BAFO: percentage of bone (area surrounded by yellow marking) in the healing chamber (area surrounded by green marking).

**Figure 5 materials-15-00496-f005:**
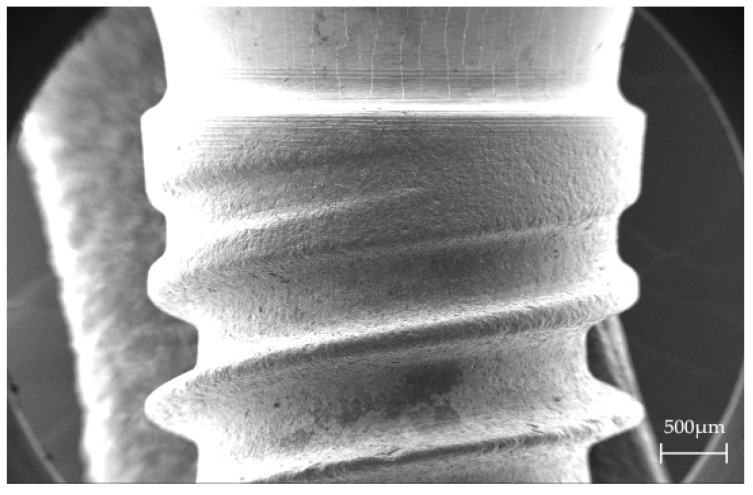
SEM-Image of a whiteSKY zirconia implant, 45-fold magnification.

**Figure 6 materials-15-00496-f006:**
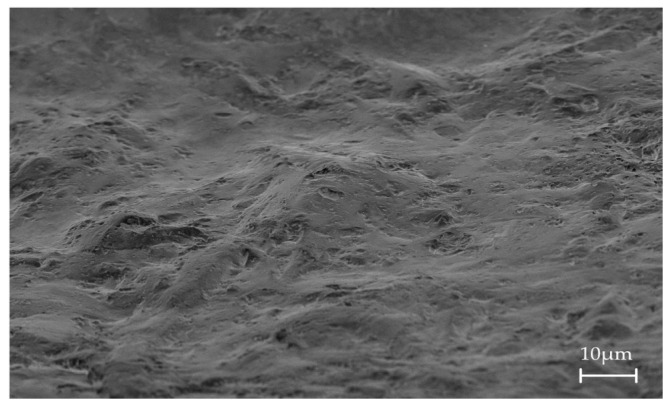
SEM-Image of a whiteSKY zirconia implant, 2000-fold magnification.

**Figure 7 materials-15-00496-f007:**
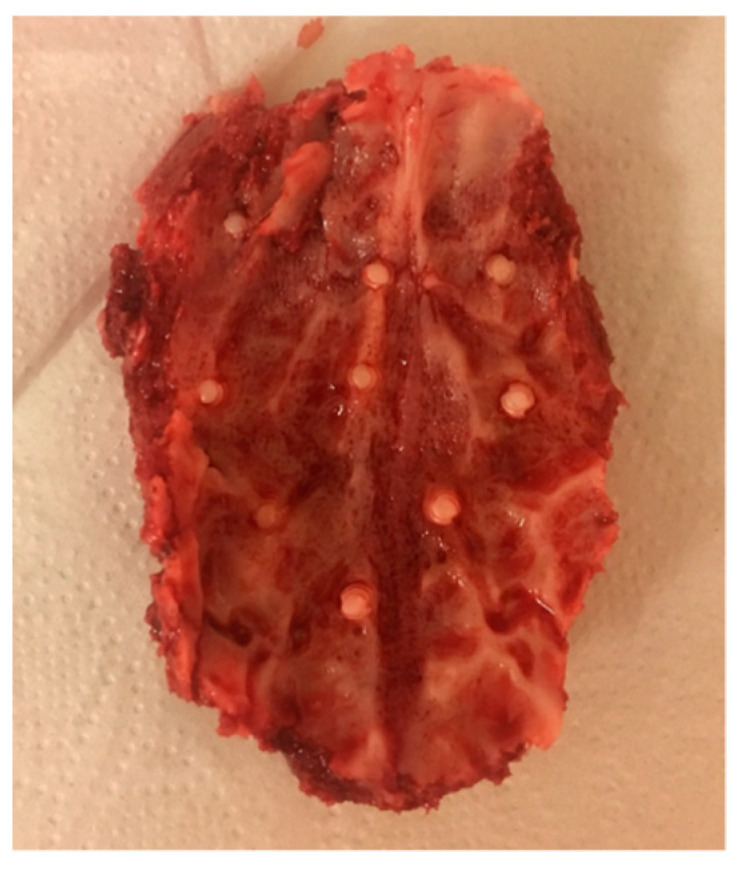
Extracted frontal calvarial bone of a landrace pig after a healing time of four weeks.

**Figure 8 materials-15-00496-f008:**
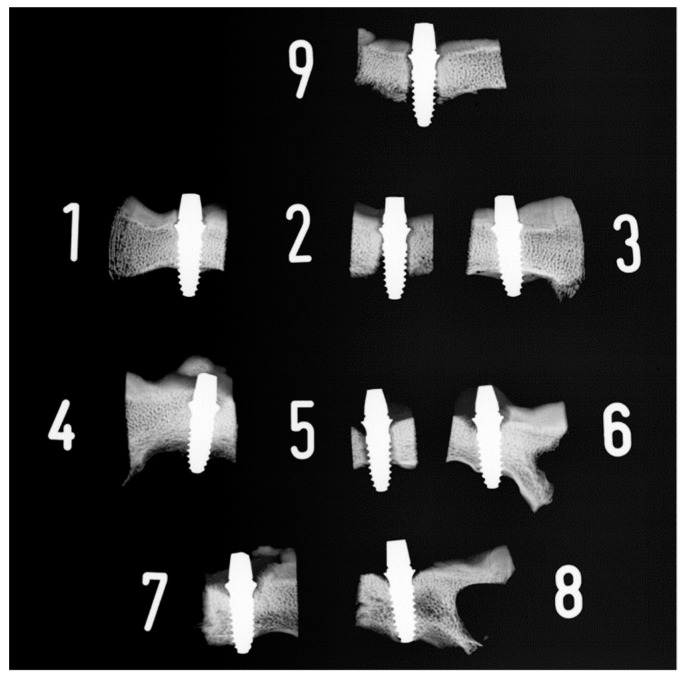
Radiographs of all dental implants that were placed in animal 3 in the frontal calvarial bone after four weeks of healing. Implants 2, 5, and 9 are radiographically not osseointegrated, presenting a peri-implant brightening. Implants 1 and 6–8 are radiographically fully osseointegrated and partially overgrown by bone.

**Figure 9 materials-15-00496-f009:**
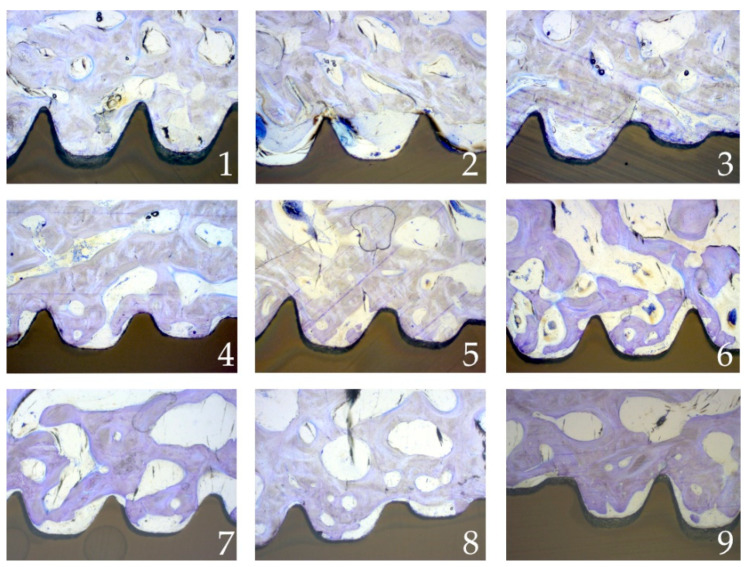
Microscope images after healing and histomorphometrical preparation. Upper row after 2 weeks of healing (1. Control, 2. CAP, 3. UV-light), middle row after 4 weeks of healing (4. Control, 5. CAP, 6. UV-light), lower row after 9 weeks of healing (7. Control, 8. CAP, 9. UV-light).

**Figure 10 materials-15-00496-f010:**
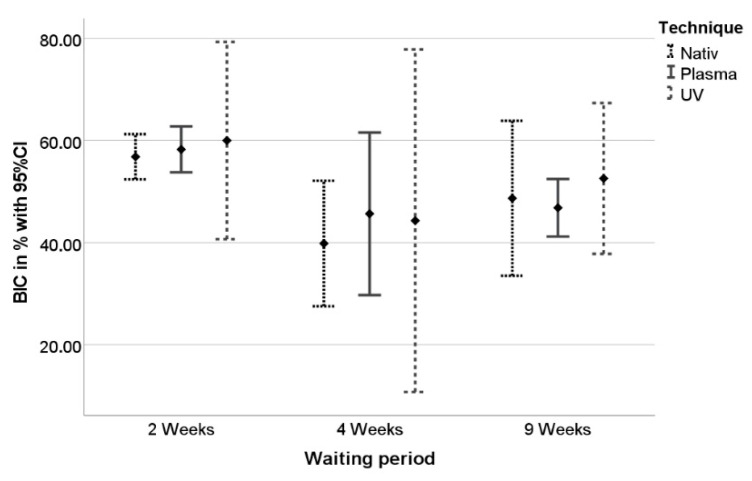
BIC values in percent ± standard deviation after 2, 4, and 9 weeks of healing.

**Figure 11 materials-15-00496-f011:**
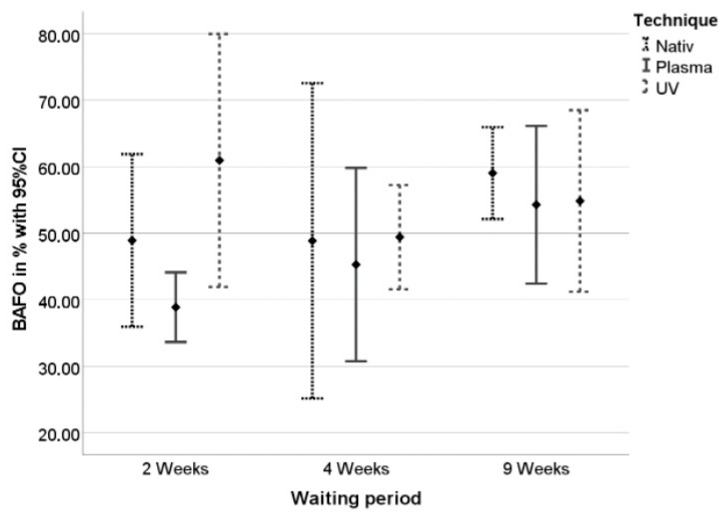
BAFO values in percent ± standard deviation after 2, 4, and 9 weeks of healing.

**Table 1 materials-15-00496-t001:** BIC values after 2, 4, and 9 weeks of healing.

Healing Period	Surface Treatment	Number (n)	BIC Mean (in %)	Min/Max (in %)
2 weeks	Control	5	56.82	51.63/61.72
	CAP	4	49.09	21.85/60.58
	UV	4	60.01	39.03/80.11
4 weeks	Control	4	36.51	26.52/47.97
	CAP	5	45.66	30.06/69.91
	UV	4	35.10	23.25/49.62
9 weeks	Control	6	44.66	24.5/70.74
	CAP	5	43.30	29.28/51.22
	UV	5	43.20	29.01/61.95

**Table 2 materials-15-00496-t002:** BAFO values after 2, 4, and 9 weeks of healing.

Healing Period	Surface Treatment	Number (n)	BAFO Mean (in %)	Min/Max (in %)
2 weeks	Control	5	48.91	35.43/70.89
	CAP	4	37.16	32.02/42.04
	UV	3	60.95	48.63/70.77
4 weeks	Control	4	47.79	32.91/59.45
	CAP	5	45.28	22.66/59.99
	UV	4	46.65	43.0/50.66
9 weeks	Control	6	59.7	50.2/68.12
	CAP	5	54.65	40.44/64.43
	UV	5	58.47	49.48/68.75

**Table 3 materials-15-00496-t003:** Statistical results of the generalized linear mixed model (GLMM) concerning differences in BIC values between the different treatment groups.

Test	2 Weeks	4 Weeks	9 Weeks
UV/Control	0.6890	0.6798	0.6537
CAP/Control	0.8678	0.5031	0.8149
UV/CAP	0.8470	0.8916	0.5268

**Table 4 materials-15-00496-t004:** Statistical results of the generalized linear mixed model (GLMM) concerning differences in BIC values between the different time points.

Test	2 Weeks	4 Weeks	9 Weeks
2 weeks/4 weeks	0.595	0.1357	0.1546
4 weeks/9 weeks	0.3118	0.4472	0.8829
2 weeks/9 weeks	0.2838	0.4149	0.2143

**Table 5 materials-15-00496-t005:** Characterization of non-osseointegrated implants during the different healing times.

Implant Number	Animal	Healing Period (weeks)	Surface Treatment	BIC	BAFO
5	1	2	CAP	1.26	4.19
9	1	2	Control	4.05	0
11	2	2	CAP	18.86	38.25
14	2	2	UV	1.04	10.36
18	2	2	UV	0	0.81
20	3	4	CAP	0	0
23	3	4	Control	0	0
27	3	4	UV	0	0.63
28	4	4	UV	3.05	12.5
29	4	4	Control	7.93	17.02
45	5	9	UV	0	0
54	6	9	CAP	0	0

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
