# Peer review of "Osseointegration of Zirconia Implants after UV-Light or Cold Atmospheric Plasma Surface Treatment In Vivo"

_materials, 2022, doi:10.3390/ma15020496_

Round 1

Reviewer 1 Report

The paper entitled “Osseointegration of zirconia implants after UV-light or cold atmospheric plasma surface treatment in vivo” is a contribute that evaluate the effects of UV-light and cold plasma 57on osseointegration of yttria-stabilized zirconia implants in an vivo model.

The paper can be of interest to the reader, however some corrections are required before it can be considered suitable for publication

INTRODUCTION

The introduction part must be revised because too synthetic; the importance of the research is not noted, the scientific background relating to the topic of the study is not adequately described and also the hypothesis of the study is not mentioned . Authors should better detail the characteristics and the differences between the surface treatment using ultraviolet (UV) light or cold atmospheric plasma (CAP). A more detailed description must also be made relatively bone-to-implant contact (BIC) method used to evaluate bone apposition on implant surfaces

MATERIAL AND METHODS

All aspects relating to the scientific methodology used were described in a clear and exhaustive manner.

The design of the study, the methods and types of outcomes measured, the type of statistical analysis carried out for the interpretation of the data is correct and clearly reported.

Does the number of guinea pigs included in the study satisfy the power analysis to demonstrate the validity of the results obtained?

The iconography could be improved with more representative images. Microscope images of the histological slices must be inserted for each group of study.

RESULTS

The results are described in a precise and detailed manner; the graphical representation is well executed and allows a faster understanding of the results achieved in the study. The footnotes of the figures clearly describe the characteristics of the image to which they refer.

DISCUSSION

The discussion of the results is on the whole well articulated ; the comparison with other studies in the literature is exhaustive. However, the significance of the results obtained in the study and the methodological limitations of the research should be emphasized

CONCLUSION

A point by point rappresentation of the conclusion part could be more effective.

Author Response

The paper entitled “Osseointegration of zirconia implants after UV-light or cold atmospheric plasma surface treatment in vivo” is a contribute that evaluate the effects of UV-light and cold plasma on osseointegration of yttria-stabilized zirconia implants in an vivo model.

The paper can be of interest to the reader, however some corrections are required before it can be considered suitable for publication

INTRODUCTION

The introduction part must be revised because too synthetic; the importance of the research is not noted, the scientific background relating to the topic of the study is not adequately described and also the hypothesis of the study is not mentioned . Authors should better detail the characteristics and the differences between the surface treatment using ultraviolet (UV) light or cold atmospheric plasma (CAP). A more detailed description must also be made relatively bone-to-implant contact (BIC) method used to evaluate bone apposition on implant surfaces

Answer: We revised the introduction according to your advice and added a detailed description of the surface treatments we used. We added our hypothesis and extended the scientific background relating to the topic of the study.

MATERIAL AND METHODS

All aspects relating to the scientific methodology used were described in a clear and exhaustive manner.

The design of the study, the methods and types of outcomes measured, the type of statistical analysis carried out for the interpretation of the data is correct and clearly reported.

Does the number of guinea pigs included in the study satisfy the power analysis to demonstrate the validity of the results obtained?

Answer: According to our statistician, theoretically, the number of pigs would be irrelevant, since the statistical test relates the effect to the variance and the number of pigs resulting in the p-value. If, one would reduce the number of the pigs with the same effect and the same variance, the p-value would be drastically worse. A significant p-value states that the number of individuals is sufficient to statistically prove the effect.

The iconography could be improved with more representative images. Microscope images of the histological slices must be inserted for each group of study.

Answer: We added several representative images as well as microscope images.

RESULTS

The results are described in a precise and detailed manner; the graphical representation is well executed and allows a faster understanding of the results achieved in the study. The footnotes of the figures clearly describe the characteristics of the image to which they refer.

DISCUSSION

The discussion of the results is on the whole well articulated ; the comparison with other studies in the literature is exhaustive. However, the significance of the results obtained in the study and the methodological limitations of the research should be emphasized

Answer: We revised the Discussion section according to your advice. We have emphasized on the significance of the results obtained and added two paragraphs on the methodological limitations of the study.

CONCLUSION

A point by point rappresentation of the conclusion part could be more effective.

Answer: We revised the conclusion section according to your advice.

Reviewer 2 Report

In this study, the surface of yttria-stabilized zirconia implant were UV-light or cold atmospheric plasma-treated, and the bone-implant contact rate and bone area fraction occupancy (bone implant contact values within the retentive parts of the implants) were examined for the effect on the bone around the implant.

I recommend that this paper be accepted after revision. The focus of the research is not bad, but the data is scarce and needs to be added.

Major Comments:

Materials and methods

2.1 Implant

Please attach a image of the implant so that you can see the shape of the implant.

Results:

The surface treatment of the implant was performed, but there are no images of the surfaces. Please add the results (surface texture) of each surface treatment (SEM images, something surface analysis). I think it is necessary to consider the results of BIC and BAFO.

Is there any presentation of specimens? You prepared macroscopic and radiological evaluation but there is no indication of the results. It is interesting to observe if there are histological and radiological difference in the peri-implant bone of each group.

It is a pity that there is no result on surface treatment, despite studies looking at the effects of implant surface modification. I strongly hope to add the data there.

3.3 Histomorphometrical evaluation

Line 179, 186, 189 etc

Isn’t the observation period in this experiment 2 weeks, 4 weeks and 9 weeks? In figure 2 and 3, it is 9 weeks, but in the text (abstract and line 179, 186, 189 etc) it is 8 weeks. Which is correct?

Minor Comments:

  1. Discussion

Line 210

Please insert [6] before period.

There are spaces between letters here and there, so make sure to do the final check properly.

Author Response

In this study, the surface of yttria-stabilized zirconia implant were UV-light or cold atmospheric plasma-treated, and the bone-implant contact rate and bone area fraction occupancy (bone implant contact values within the retentive parts of the implants) were examined for the effect on the bone around the implant.

I recommend that this paper be accepted after revision. The focus of the research is not bad, but the data is scarce and needs to be added.

Major Comments:

Materials and methods

2.1 Implant

Please attach a image of the implant so that you can see the shape of the implant.

Answer: We added an image of the implant in chapter 2.1.

Results:

The surface treatment of the implant was performed, but there are no images of the surfaces. Please add the results (surface texture) of each surface treatment (SEM images, something surface analysis). I think it is necessary to consider the results of BIC and BAFO.

Answer: We added 2 SEM images of the surface structure of the implants that were used. We did not find any statistical significant differences in BIC and BAFO after the surface treatments in this study. In previous in vitro studies we did not find any changes in surface characteristics (SEM and surface roughness) after treatment using UV light or cold atmospheric plasma on titanium or zirconia surfaces similar to those used in this study (please see Smeets R, Henningsen A, Heuberger R, Hanisch O, Schwarz F, Precht C. Influence of UV Irradiation and Cold Atmospheric Pressure Plasma on Zirconia Surfaces: An In Vitro Study. Int J Oral Maxillofac Implants. 2019 March/April;34(2):329–336. doi: 10.11607/jomi.7017; Henningsen A, Smeets R, Heuberger R, Jung OT, Hanken H, Heiland M, Cacaci C, Precht C. Changes in surface characteristics of titanium and zirconia after surface treatment with ultraviolet light or non-thermal plasma. Eur J Oral Sci. 2018 Apr;126(2):126-134. doi: 10.1111/eos.12400 and Henningsen A, Smeets R, Hartjen P, Heinrich O, Heuberger R, Heiland M, Precht C, Cacaci C. Photofunctionalization and non-thermal plasma activation of titanium surfaces. Clin Oral Investig. 2018 Mar;22(2):1045-1054. doi: 10.1007/s00784-017-2186-z). Therefore we did not investigate the changes in surface roughness or surface topography. However, we added a SEM image to show the surface characteristic of an implant.

Is there any presentation of specimens? You prepared macroscopic and radiological evaluation but there is no indication of the results. It is interesting to observe if there are histological and radiological difference in the peri-implant bone of each group.

Answer: We added several tables to show the differences in BIC and BAFO between the groups. Additionally, we added figure 9 to give examples of the histomorphometrical results. However, we did not found histological differences between the groups.

It is a pity that there is no result on surface treatment, despite studies looking at the effects of implant surface modification. I strongly hope to add the data there.

Answer: Dear reviewer, we are with you! We were also very surprised that there was no effect in this study. However, we checked the results several times, but there was no statistically significant effect. We strongly believe that the animal model could be a reason. The animals gained 68% body weight in only 9 weeks. We added the sentence “This may also be the reason that there were no statistically significant differences on BIC and BAFO between treated and non-treated implants in this study.” To the limitations section.

3.3 Histomorphometrical evaluation

Line 179, 186, 189 etc

Isn’t the observation period in this experiment 2 weeks, 4 weeks and 9 weeks? In figure 2 and 3, it is 9 weeks, but in the text (abstract and line 179, 186, 189 etc) it is 8 weeks. Which is correct?

Answer: This is absolutely correct. We had 2 weeks, 4 weeks and 9 weeks as observation period and corrected the mistakes.

Minor Comments:

  1. Discussion

Line 210

Please insert [6] before period.

Answer: we corrected the mistake.

There are spaces between letters here and there, so make sure to do the final check properly.

Answer: We did a thorough final check and deleted the spaces.

Reviewer 3 Report

Dear Authors,

This is a good study and well written manuscript!

Abstract:

  1. The first sentence please check English sentence structure. "in a comparison of the two methods" looks 
  2. The second sentence: Influence on the "osseointegration" of dental zr implants in an animal model.

Method:

1. The reason of use parietal bone in this study and its limitation

2. Sample size calculation. This study has 6 implants per group at a specific time point.

3. Any reference/evidence support for the surface treatment protocol

4. A study flowchart may help the readers to understand the number of samples and implants in each treatment groups and in each time points.

5. What measures authors did to ensure the sample blocks are parallel? How did the authors make sure the samples are sectioned at the same/similar plane? 

6. Any figures of the microscope images with and without measurements

7. Who perform the measurements? By human (how many operators) or by software? Reliable and repeatable?

8. Regarding the statistical analysis, would the authors also provides information of statistics performed between the groups.

Results:

  1. 74% implants were partially inserted into the sinus. Were these intentionally or accidentally? Would this affect the main outcome (bone contact) of this study?
  2. Would authors also presenting the numeric value of the BIC values and statistical tests performed in a table?
  3. Fig 2, the SD values looks big especially at 2 weeks. However, we may need to see the numeric result to comment further. We may need to see if any analyzing statistics performed in figure 2 and 3.

Author Response

Dear Authors,

This is a good study and well written manuscript!

Abstract:

  1. The first sentence please check English sentence structure. "in a comparison of the two methods" looks 

  1. The second sentence: Influence on the "osseointegration" of dental zr implants in an animal model.

Answer: Dear Reviewer, thank you very much! We corrected the two sentences.

Method:

  1. The reason of use parietal bone in this study and its limitation

Answer: A frontal calvarial bone model was used because it enables to place up to 9 implants using only one surgical approach in a short time in order to reduce the stress on the animals. We added this to the M&M section. We also added the limitations in the discussion section.

  1. Sample size calculation. This study has 6 implants per group at a specific time point.

Answer: We added the sample size calculation in chapter 2.3.

  1. Any reference/evidence support for the surface treatment protocol

Answer: We added the references for the surface treatment protocol in chapter 2.2

  1. A study flowchart may help the readers to understand the number of samples and implants in each treatment groups and in each time points.

Answer: We added a study flowchart in chapter 2.2.

  1. What measures authors did to ensure the sample blocks are parallel? How did the authors make sure the samples are sectioned at the same/similar plane? 

Answer: There are no sample blocks, but preparatory blocks. The samples were embedded in plastic and all implants were histologically processed. The longitudinal axis of the implants was taken as the grinding plane. It is to be regarded as representative. The parallelism of the blocks is a basic requirement for the preparation of the section. The area with the largest diameter was targeted, the fluctuation range is around ± 200 µm.

  1. Any figures of the microscope images with and without measurements

Answer: We added a microscope images after 2, 4 and 9 weeks of healing in chapter 3.3. We try to explain the measurement of BIC and BAFO with images in chapter 2.5.

  1. Who perform the measurements? By human (how many operators) or by software? Reliable and repeatable?

Answer: Measurements were performed using the Osteomeasure XP® software by two independent resarchers. We added the inter-observer reliability.

  1. Regarding the statistical analysis, would the authors also provide information of statistics performed between the groups.

Answer: We added the statistical results.

Results:

  1. 74% implants were partially inserted into the sinus. Were these intentionally or accidentally? Would this affect the main outcome (bone contact) of this study?

Answer: The insertions into the sinus were accidentally due to the limited amount of bone in the pigs. We added this to the section. The main outcome was not affected because we only counted the parts of the implants surface covered with mineralized bone tissue and placed it in relation to the total surface giving the percentage of the surface that was successfully osseointegrated.

  1. Would authors also presenting the numeric value of the BIC values and statistical tests performed in a table?

Answer: We added the statistical results in tables 1 and 2.

  1. Fig 2, the SD values looks big especially at 2 weeks. However, we may need to see the numeric result to comment further. We may need to see if any analyzing statistics performed in figure 2 and 3.

Answer: We noticed high SD values especially at 2 weeks. The low number of animals and implants may be possible reasons. However, another reason may be that the process of osseointegration may be different compared to the osseointegration process of titanium implants. We added a section to the limitations in the discussion section.

Round 2

Reviewer 2 Report

I'm sorry for the late reply. The content has improved a lot. The specimens that I thought were necessary were accurately placed.   I recommend that this paper be accepted.